# Debugging and Explaining Metric Learning Approaches: An Influence Function Based Perspective

**Ruofan Liu**
Shanghai Jiao Tong University
National University of Singapore
`liu.ruofan16@u.nus.edu`

**Yun Lin**[*]
Shanghai Jiao Tong University
National University of Singapore
`llmhyy@gmail.com`

**Xianglin Yang**
National University of Singapore
`xianglin@u.nus.edu`

**Jin Song Dong**
National University of Singapore
`dcsdjs@nus.edu.sg`

## Abstract

Deep metric learning (DML) learns a generalizable embedding space where the representations of semantically similar samples are closer. Despite achieving good performance, the state-of-the-art models still suffer from the generalization errors such as farther similar samples and closer dissimilar samples in the space. In this work, we design empirical influence function (EIF), a debugging and explaining technique for the generalization errors of the state-of-the-art metric learning models. EIF is designed to efficiently identify and quantify how a subset of training samples contribute to the generalization errors. Moreover, given a user-specific error, EIF can be used to relabel a potentially noisy training sample as a mitigation. In our quantitative experiment, EIF outperforms the traditional baseline in identifying more relevant training samples with statistical significance and 33.5% less time. In the field study on the well-known datasets such as CUB200, CARS196, and InShop, EIF identifies 4.4%, 6.6%, and 17.7% labelling mistakes, indicating the direction of the DML community to further improve the model performance. Our code is available at `https://github.com/lindsey98/Influence_function_metric_learning`.

## 1 Introduction

Deep metric learning (DML) learns a generalizable embedding space of a dataset, where semantically similar samples are mapped closer [12]. It has been widely applied in face recognition [27], image retrieval [38], and clustering [5]. Recently, the record-breaking methodologies have been generally evolving from pairwise-based approaches (e.g., triplet-based [12] and pair-based [11]) to proxy-based approaches [20, 14, 31, 10, 23]. However, many recent works [28, 25] begin to achieve only marginal improvements on the classical datasets [37, 18, 19]. Thus, the explanation approaches of DML are in need for understanding *why the trained model confuses the dissimilar samples and fails to recognize the similar samples*.

This research starts with our investigation on popular classical datasets (i.e., CUB200, CARS196, and InShop) that, for state-of-the-art metric learning approaches, (1) different approaches share not only similar performance metrics (e.g., Recall@1), but the same types of generalization errors, and (2) the human inspection sometimes has no less generalization errors on existing DML datasets. We

---

[*]Corresponding author

36th Conference on Neural Information Processing Systems (NeurIPS 2022).

report the results at Section 5 and our website [2]. The observation leads us to design an influence function based explanation framework to investigate the existing datasets, consisting of:

- **Scalable training-sample attribution:** We propose empirical influential function (EIF) to (1) identify what training samples contribute to the generalization errors, and (2) quantify how much contribution they make to the errors. Technically, we replace the Hessian matrix in traditional influence function [15] with light-weighted Newton step estimation to improve both its effectiveness and efficiency.

- **Dataset relabelling recommendation:** We further aim to identify the potentially "buggy" training samples with mistaken labels and generate their relabelling recommendation.

**Technique Evaluation.** We design one sample-locating experiment and one sample-relabelling experiment on 3 datasets to evaluate our framework. In the sample-locating experiment, we evaluate EIF's performance on locating the influential training samples for both individual-pair confusion (i.e., model is confused on one pair of samples) and group-pair confusion (i.e., model is confused on a class of samples). The results show that, compared to the traditional influence function [15], EIF identifies root-cause training samples which can significantly mitigate the confusion distances with 33.5% less time. As for the sample-relabelling experiment, we inject 10% noisy training samples to train a DML model. Our results show that we can accurately recommend on average 88.2% of the mis-labelled training samples.

**Empirical Investigation.** Based on the proposed framework, we investigate the classical datasets such as CUB200 [37], CARS196 [18], and InShop [19]. We find that the labels annotated in the datasets are more unreliable than expected. We summarize a taxonomy of dataset problems (e.g., see Section 5), We further conclude that the major barriers for DML performance might not be the model design, but the confusing labels in the classical datasets.

In summary, this work makes the following contributions:

- We propose empirical influence function (EIF) for DML approaches, which can attribute root-cause training samples for arbitrary number of pairs of unseen test samples.

- We propose a sample-relabelling technique based on EIF for mitigating potential dataset problems.

- We identify and categorize labelling problems of the well-known classical datasets for DML, indicating the potential direction to further improve the performance of DML approaches.

## 2 Problem Setting

We denote the input space as $\mathcal{X} \subset \mathbb{R}^d$ ($d$ is the input dimension), the embedding space as $\mathcal{Z} \subset \mathbb{R}^m$ ($m$ is the embedding dimension), and the class label space $\mathcal{Y} \subset Z^+$. A DML network $f(.)$ parameterized by $\boldsymbol{\theta}$ is denoted as $f_{\boldsymbol{\theta}} : \mathcal{X} \to \mathcal{Z}$. Given a distance measure $d(.,.)$ where $d : \mathcal{Z} \times \mathcal{Z} \to \{0, \mathbb{R}^+\}$, we can calculate the distance between any input pairs $(\mathbf{x}_i, \mathbf{x}_j) \in \mathcal{X}$ by $d(f_{\boldsymbol{\theta}}(\mathbf{x}_i), f_{\boldsymbol{\theta}}(\mathbf{x}_j))$. Typically, cosine distance and Euclidean distance are common choices of $d(.,.)$. In addition, we define a labelling function $\mathcal{R}(.)$ where $\mathcal{R} : \mathcal{X} \to \mathcal{Y}$.

The DML techniques aim to optimize $\boldsymbol{\theta}$ such that $\forall \mathbf{x}_i, \mathbf{x}_j, \mathbf{x}_k \in \mathcal{X}$ ($y_i, y_j, y_k \in \mathcal{Y}$, $y_i = y_j \neq y_k$), $d(f_{\boldsymbol{\theta}}(\mathbf{x}_i), f_{\boldsymbol{\theta}}(\mathbf{x}_j)) < d(f_{\boldsymbol{\theta}}(\mathbf{x}_i), f_{\boldsymbol{\theta}}(\mathbf{x}_k))$. Pair-based losses try to optimize the inequality directly on pairs / triplets [11, 12, 29]. However, studies have shown that the optimization suffer from (1) slow and noisy convergence, (2) high computational complexity. Therefore, proxy-based losses have been proposed to address these issues [20, 31, 14, 23, 6], which significantly outperform the pair-based losses. Most of proxy-based losses are defined on a per-sample basis [20, 31, 23, 6]. Therefore, in this work, we follow the loss of form $\mathcal{L}(\mathbf{x}; \boldsymbol{\theta})$.

Given the training dataset set $\mathcal{X}_{train}$ with labels $\mathcal{Y}_{train}$ and the testing dataset $\mathcal{X}_{test}$ with labels $\mathcal{Y}_{test}$ where training and testing are class disjoint, i.e., $\mathcal{Y}_{train} \cap \mathcal{Y}_{test} = \emptyset$. The generalization error can be defined as a testing sample not sharing the same class label as its nearest neighbor in the space, i.e.,

**Definition 2.1.** We define a testing-sample pair $p = (\mathbf{x}_i, \mathbf{x}_j)$ ($\mathbf{x}_i, \mathbf{x}_j \in \mathcal{X}_{test}$) as a **confusion pair** if:

1. $\mathbf{z}_j = f_{\boldsymbol{\theta}}(\mathbf{x}_j)$ is the nearest neighbour of $\mathbf{z}_i = f_{\boldsymbol{\theta}}(\mathbf{x}_i)$ in the distance space;

2. $y_i \neq y_j$ and $y_i, y_j \in \mathcal{Y}_{test}$.

Given a set of confusion pairs $P_c = \{p_1, p_2, ..., p_n\}$, we aim to achieve the following two goals:

**G1. Influential Sample Identification**   We aim to locate the set of root-cause training samples $\mathcal{X}_r \subset \mathcal{X}_{train}$ such that retraining with re-weighted $\mathcal{X}_r$ can increase the average distance of confusion pairs in $P_c$ the most,

$$
\begin{aligned}
\mathcal{X}_r &= \underset{\mathcal{X}_r \subset \mathcal{X}_{train}}{\arg\max} \frac{1}{|P_c|} \sum_{(\mathbf{x}_i, \mathbf{x}_j) \in P_c} d(\mathbf{x}_i, \mathbf{x}_j; \hat{\boldsymbol{\theta}}_r) \textit{ subject to} \\
\hat{\boldsymbol{\theta}}_r &= \underset{\boldsymbol{\theta}}{\arg\min} \frac{1}{|\mathcal{X}_{train}|} \Big[ \sum_{\mathbf{x_i} \in \mathcal{X}_{train} \setminus \mathcal{X}_r} \mathcal{L}(\mathbf{x_i}, y_i; \boldsymbol{\theta}) + \sum_{\mathbf{x_i} \in \mathcal{X}_r} \epsilon_i \mathcal{L}(\mathbf{x_i}, y_i; \boldsymbol{\theta}) \Big]
\end{aligned}
\tag{1}
$$

In Equation 1, $\hat{\boldsymbol{\theta}}_r$ is the retrained model by re-weighting the training sample $\mathbf{x}_i$ by $\epsilon_i$. Specifically, $|\epsilon_i|$ is the re-weighting magnitude, if $\mathbf{x_i}$ is a helpful training sample (i.e. helpful in de-confusion), then $\epsilon_i$ is set to be greater than one; otherwise for a harmful training, $\epsilon_i$ is set to be less than one. Under certain choice of $\mathcal{X}_r$, the average distance of the confusion pair set $P_c$ is maximized.

**G2. Influential Sample Relabelling**   We aim to find the set of root-cause training samples $\mathcal{X}_l \subset \mathcal{X}_{train}$ and a relabelling function $\mathcal{R} : \mathcal{X} \to \mathcal{Y}$ such that retraining $\boldsymbol{\theta}$ by changing the labels of $\mathcal{X}_l$ with $\mathcal{R}(.)$ can increase the average distance of confusion pairs in $P_c$ the most. Specifically,

$$
\begin{aligned}
\mathcal{X}_l &= \underset{\mathcal{X}_l \subset \mathcal{X}_{train}, \mathcal{R}(.)}{\arg\max} \frac{1}{|P_c|} \sum_{(\mathbf{x}_i, \mathbf{x}_j) \in P_c} d(\mathbf{x}_i, \mathbf{x}_j; \hat{\boldsymbol{\theta}}_l) \textit{ subject to} \\
\hat{\boldsymbol{\theta}}_l &= \underset{\boldsymbol{\theta}}{\arg\min} \frac{1}{|\mathcal{X}_{train}|} \Big[ \sum_{\mathbf{x}_i \in \mathcal{X}_{train} \setminus \mathcal{X}_l} \mathcal{L}(\mathbf{x}_i, y_i; \boldsymbol{\theta}) + \sum_{\mathbf{x}_i \in \mathcal{X}_l} \mathcal{L}(\mathbf{x}_i, \mathcal{R}(\mathbf{x}_i); \boldsymbol{\theta}) \Big]
\end{aligned}
\tag{2}
$$

## 3   Approach

**Recaping Influence Function**   Given that a learned model parameterized by $\hat{\boldsymbol{\theta}}$, and its DML loss $\mathcal{L}(\mathbf{x}; \hat{\boldsymbol{\theta}})$. The influence of up-weighting a training sample $\mathbf{x}_{train}$ on a testing sample $\mathbf{x}_{test}$ is [15]:

$$
\mathcal{I}(\mathbf{x}_{train}, \mathbf{x}_{test}) = \mathcal{L}(\mathbf{x}_{test}; \hat{\boldsymbol{\theta}}') - \mathcal{L}(\mathbf{x}_{test}; \hat{\boldsymbol{\theta}}) \approx -\nabla_{\hat{\boldsymbol{\theta}}} \mathcal{L}(\mathbf{x}_{test}; \hat{\boldsymbol{\theta}})^\top \mathcal{H}_{\hat{\boldsymbol{\theta}}}^{-1} \nabla_{\hat{\boldsymbol{\theta}}} \mathcal{L}(\mathbf{x}_{train}; \hat{\boldsymbol{\theta}})
\tag{3}
$$

In Equation 3, $\hat{\boldsymbol{\theta}}$ is the original model, $\hat{\boldsymbol{\theta}}'$ is the retrained model after $\mathbf{x}_{train}$ is adjusted, $\mathcal{H}_{\boldsymbol{\theta}}^{-1} = \frac{1}{|\mathcal{X}_{train}|} \sum_{i=1}^{|\mathcal{X}_{train}|} \nabla_{\boldsymbol{\theta}}^2 \mathcal{L}(\mathbf{x}_i, \hat{\boldsymbol{\theta}})$. The first term can be interpreted as the testing sample's sensitivity to the model's parameters, the second term estimates the interaction between training samples, and the third term is the influence of this training sample to the model. In DML settings, testing loss is undefined according to Equation 3 since testing classes are unseen. Given a confusion pair $p_c = (\mathbf{x}_i, \mathbf{x}_j)$ with distance $d(p_c; \hat{\boldsymbol{\theta}})$, we can replace the testing loss with $d(p_c; \hat{\boldsymbol{\theta}})$, i.e.,

$$
\mathcal{I}(\mathbf{x}_{train}, p_c) = -\nabla_{\hat{\boldsymbol{\theta}}} d(p_c; \hat{\boldsymbol{\theta}})^\top \mathcal{H}_{\hat{\boldsymbol{\theta}}}^{-1} \nabla_{\hat{\boldsymbol{\theta}}} \mathcal{L}(\mathbf{x}_{train}; \hat{\boldsymbol{\theta}})
\tag{4}
$$

However, computing $\mathcal{I}(\mathbf{x}_{train}, p_c)$ still suffers from two drawbacks, i.e., (1) high computational cost and (2) inaccurate approximation for group-pair confusion.

**High Computational Cost**   Computing the Hessian function $\mathcal{H}_{\hat{\boldsymbol{\theta}}}^{-1}$ is non-trivial, which requires the complexity of $\mathcal{O}(np^2 + p^3)$ where $n$ is training dataset size and $p$ is the parameter size. In [15], the complexity is further reduced to $\mathcal{O}(np + rtp)$ where $rt \sim \mathcal{O}(n)$. With the increase of parameter size (e.g., millions) and the training set, the runtime cost is still considerably large.

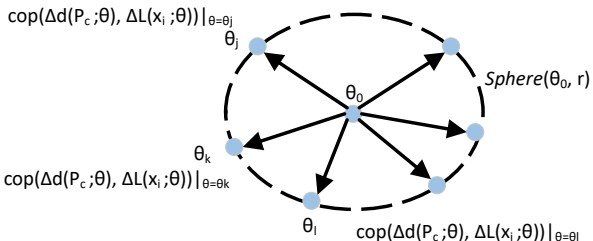

Figure 1: Illustration of EIF calculation by sampling most representative $\boldsymbol{\theta}^{(m)}$ on the hypersphere.

**Accuracy of Influence of Group-pair Confusion**  When we are locating the influential training samples for a group of confusion pairs, the assumption of Equation 4 cannot hold in practice because the derivation of the Hessian matrix $\mathcal{H}_{\hat{\boldsymbol{\theta}}}^{-1}$ in Equation 3 requires that $\hat{\boldsymbol{\theta}}' \sim \hat{\boldsymbol{\theta}}$ so that Taylor expansion can be applied [15]. However, the number of influential training samples to a group of confusion pairs (i.e., $\mathcal{X}_r$ in Equation 1) can be large, indicating that the adjusted parameters $\hat{\boldsymbol{\theta}}'$ could be very different from the original $\hat{\boldsymbol{\theta}}$. Thus, such approximation can lead to inaccurate estimation.

**Empirical Influence Function (EIF)**  In this work, we design empirical influence function (EIF) to address the above challenges. Our rationale lies in that, (1) fitting harmful samples contribute to more generalization errors, and (2) fitting helpful samples contribute to mitigating the generalization errors. We estimate the influence scores by the empirical co-change product between the distance of confusion pairs $\Delta d(P_c; \boldsymbol{\theta})$ and the training loss $\Delta\mathcal{L}(\mathbf{x}; \boldsymbol{\theta})$ ($\mathbf{x} \in \mathcal{X}_{train}$) as Equation 5:

$$\mathcal{I}(\mathbf{x}, P_c) = E_{\boldsymbol{\theta}}[cop(\Delta d(P_c; \boldsymbol{\theta}), \Delta\mathcal{L}(\mathbf{x}; \boldsymbol{\theta}))] = \int \frac{\Delta d(P_c; \boldsymbol{\theta})\Delta\mathcal{L}(\mathbf{x}; \boldsymbol{\theta})}{||\boldsymbol{\theta} - \boldsymbol{\theta}_0||_2}p(\boldsymbol{\theta})d\boldsymbol{\theta} \qquad (5)$$

Assume that we train the network by upweighting $\mathbf{x}$ to have $\boldsymbol{\theta}_0 \to \boldsymbol{\theta}$, the co-change product is defined as $cop(\Delta d(P_c; \boldsymbol{\theta}), \Delta\mathcal{L}(\mathbf{x}; \boldsymbol{\theta})) = \frac{\Delta d(P_c; \boldsymbol{\theta})\Delta\mathcal{L}(\mathbf{x}; \boldsymbol{\theta})}{||\boldsymbol{\theta} - \boldsymbol{\theta}_0||_2}$. $cop(\Delta d(P_c; \boldsymbol{\theta}), \Delta\mathcal{L}(\mathbf{x}; \boldsymbol{\theta})) > 0$ indicates that $\mathbf{x}$ is harmful. Intuitively, $\mathbf{x}$ is harmful to distinguish $P_c$ if (1) more fitting on $\mathbf{x}$ (with $\Delta\mathcal{L}(\mathbf{x}; \boldsymbol{\theta})) < 0$ co-happens with more confused pairs (with $\Delta d(P_c; \boldsymbol{\theta}) < 0$) or (2) less fitting on $\mathbf{x}$ (with $\Delta\mathcal{L}(\mathbf{x}; \boldsymbol{\theta})) > 0$ co-happens with less confused pairs (with $\Delta d(P_c; \boldsymbol{\theta}) > 0$). On the other hand, $cop(\Delta d(P_c; \boldsymbol{\theta}), \Delta\mathcal{L}(\mathbf{x}; \boldsymbol{\theta})) < 0$ indicates that $\mathbf{x}$ is helpful. However, retraining each individual sample is prohibitively expensive, therefore we need to estimate the integral by carefully sampling representative $\boldsymbol{\theta}$'s only.

As shown in Figure 1, assume that $\boldsymbol{\theta}_0$ is the model parameter, $\boldsymbol{\theta}$ is the parameter after re-retraining, the distribution of $\boldsymbol{\theta}$ forms a high-dimensional Sphere $p(\boldsymbol{\theta}) \sim Sphere(\boldsymbol{\theta}_0, r)$ with a radius $r$, where $r$ is some pre-defined bound to limit the perturbation step. The intractable integral in Equation 5 is estimated by drawing Monte Carlo samples (e.g., $\boldsymbol{\theta}^{(1)}, \boldsymbol{\theta}^{(2)}, ... \boldsymbol{\theta}^{(m)}$) on the sphere. i.e., we sample $\Theta \subset Sphere(\boldsymbol{\theta}_0, r)$

$$E_{\boldsymbol{\theta}}[cop(\Delta d(P_c; \boldsymbol{\theta}), \Delta\mathcal{L}(\mathbf{x}; \boldsymbol{\theta}))] \approx \frac{1}{|\Theta|}\sum_{\boldsymbol{\theta}^{(m)}\in\Theta}\frac{\Delta d(P_c; \boldsymbol{\theta}^{(m)})\Delta\mathcal{L}(\mathbf{x}; \boldsymbol{\theta}^{(m)})}{||\boldsymbol{\theta}^{(m)} - \boldsymbol{\theta}_0||_2} \qquad (6)$$

In this work, instead of random sampling, we heuristically construct $\Theta$, regarding how sensitive each $\boldsymbol{\theta}^{(m)} \in \Theta$ is to the change of $d(P_c; \boldsymbol{\theta})$. Specifically, we first sample $\boldsymbol{\theta}_{max} \in Sphere(\boldsymbol{\theta}_0, r)$ pointing to the steepest ascent direction of $d(P_c; \boldsymbol{\theta})$, which can be considered as "repairing" the network. Symmetrically, $\boldsymbol{\theta}_{min}$ is taken as the steepest descent direction of $d(P_c; \boldsymbol{\theta})$, which can be considered as "worsening" the network. Next, given a user-defined threshold $N_{\boldsymbol{\theta}}$, we repetitively sample $\boldsymbol{\theta}^{(m)} \in Sphere(\boldsymbol{\theta}_0, r)$ which is orthogonal to $\forall \boldsymbol{\theta} \in \Theta$, until $N_{\boldsymbol{\theta}}$ exhausts. $\Theta$ is calculated once and shared across the whole training set. Our experiment shows that even using $\Theta = \{\boldsymbol{\theta}_{max}\}$ can achieve an accurate influence estimation, with well improved runtime efficiency.

The complexity of our empirical influence function on the whole training set is $\mathcal{O}(p) + \mathcal{O}(N_{train} \times N_{\boldsymbol{\theta}} \times p)$, where $p$ is the parameter size. The first term stands for the complexity of constructing $\Theta$,

Table 1: Paired t-test results for alternative hypothesis on individual confusion pairs $H_1$ : $\Delta \bar{d}_{EIF}(p_k) > \Delta \bar{d}_{IF}(p_k)$. And Wilcoxon signed-rank test results for alternative hypothesis on groups of confusion pairs $H_1 : \Delta \bar{d}_{EIF}(c_m) > \Delta \bar{d}_{IF}(c_m)$

| Individual confusion | CUB200 | | CARS196 | | InShop | |
|---|---|---|---|---|---|---|
| # confusion pairs | average improvement | p-value | average improvement | p-value | average improvement | p-value |
| 30 | 2.29E-02 | 3.47E-12 | 1.47E-02 | 8.16E-06 | 1.52E-02 | 1.96E-05 |
| 50 | 2.12E-02 | 1.35E-15 | 1.20E-02 | 1.04E-07 | 1.39E-02 | 2.98E-07 |
| 70 | 1.99E-02 | 4.02E-19 | 1.31E-02 | 3.06E-11 | 1.59E-02 | 7.76E-11 |
| 90 | 1.98E-02 | 1.88E-24 | 1.25E-02 | 1.52E-12 | 1.60E-02 | 2.10E-13 |
| 100 | 1.96E-02 | 5.25E-27 | 1.27E-02 | 2.93E-13 | 1.51E-02 | 9.27E-14 |
| Group confusion | CUB200 | | CARS196 | | InShop | |
| # confusion groups | average improvement | p-value | average improvement | p-value | average improvement | p-value |
| 10 | 2.23E-02 | 9.77E-04 | 1.87E-02 | 9.77E-04 | -7.83E-04 | 9.35E-01 |
| 15 | 2.34E-02 | 3.05E-05 | 1.95E-02 | 3.05E-05 | 6.22E-04 | 7.38E-01 |
| 20 | 2.43E-02 | 9.54E-07 | 1.83E-02 | 9.54E-07 | 4.72E-04 | 7.63E-01 |
| 25 | 2.44E-02 | 2.98E-08 | 1.72E-02 | 2.98E-08 | 2.08E-03 | 5.11E-01 |
| 30 | 2.44E-02 | 9.31E-10 | 1.68E-02 | 9.31E-10 | 3.27E-03 | 3.35E-01 |

and the second term is performing forward pass on training data with $\Theta$. As a result, we can attach each $\mathbf{x} \in \mathcal{X}_{train}$ with an influence function score $\mathcal{I}(\mathbf{x}, P_c)$, which can be used to rank $\mathcal{X}_{train}$ and select the most influential training samples.

---

**Algorithm 1** Training Sample Relabelling

**Input** : Training set $\mathcal{X}_{train}$, K
**Output :** Relabelled set $S = \{(\mathbf{x}, l)\}$

1  $\quad \mathcal{X}_{harm} = \{\mathbf{x} | \mathcal{I}(\mathbf{x}, P_c) > 0, \mathbf{x} \in \mathcal{X}_{train}\}$
2  $\quad S = \emptyset$
3  $\quad$ **for** $\mathbf{x} \in \mathcal{X}_{harm}$ **do**
4  $\quad \quad \{(\mathbf{x}_{nni}, y_{nni})\} \leftarrow \text{KNN}(f_{\hat{\boldsymbol{\theta}}}(\mathbf{x}))$
$\quad \quad P(y_{new} = l) = \frac{\sum_{i=1}^{K} exp(-d(\mathbf{x}, \mathbf{x}_{nni}))\mathbf{1}(y_{nni}=l)}{\sum_{l'} \sum_{i=1}^{K} exp(-d(\mathbf{x}, \mathbf{x}_{nni}))\mathbf{1}(y_{nni}=l')} , \forall l \in Set(\mathcal{Y}_{train})$
$\quad \quad S = S \cup \{(\mathbf{x}, \arg\max_l P(y_{new} = l))\}$

5  $\quad$ **return** $S$

---

**Sample Relabelling Recommendation** Based on the empirical influence function, we propose the data relabelling as a data cleansing strategy. Given the identified influential harmful training samples $\mathcal{X}_{harm}$ from EIF. Assuming that the noisy samples are the minority in the training dataset, we recommend a label $l \in \mathcal{Y}_{train}$ according to the labels of its neighbors through a weighted-KNN algorithm (See Algorithm 1). Intuitively, the label supported by more weighted neighbours is recommended to correct a harmful sample $\mathbf{x}$. The neighbours are weighted by their proximity to $\mathbf{x}$. Note that, we use hard label re-assignment (one-hot label) in Algorithm 1 (line 5). We can also revise it to soft label re-assignment by using the probability score $P(y_{new})$ as the calibration label.

## 4 Experiment

**Experimental Settings** In this study, we use Proxy-NCA++ [31] and SoftTriple [23] loss to train DML models with ResNet-50 model architecture on three datasets, i.e., CUB200 [37], CARS196 [18], and InShop [19]. The influential sample location capability is evaluated by a DML training experiment. The sample relabelling capability is evaluated by the noisy data relabelling experiment. In the DML training experiment, we evaluate whether retraining the model by upweighting or downweighting the reported influential samples can mitigate the confusion pairs. In the noisy data relabelling experiment, we flip 1%, 5%, and 10% of the labels in the above training datasets, and evaluate whether EIF can identify and correct those noisy samples. We choose the influence function with modified testing loss function (as in Equation 4) as our baseline. More details of training configuration can be referred on our website [2].

**DML Training Experiment** In the experiment, for each dataset, we select the top-$K$ ($K$=30, 50, 70, 90, 100) most confusing pairs in its testing dataset. For each selected confusion pair $p_k$, we

Table 2: Mislabelled Sample Recommendation

| Mislabelling Ratio | Dataset / Method | CUB200 | CARS196 | InShop |
|---|---|---|---|---|
| 1% | ProxyNCA++ | 75.68% | 84.09% | 89.74% |
| | SoftTriple | 75.00% | 79.49% | 90.00% |
| 5% | ProxyNCA++ | 82.49% | 86.84% | 90.29% |
| | SoftTriple | 86.14% | 90.05% | 87.03% |
| 10%. | ProxyNCA++ | 83.28% | 91.69% | 88.37% |
| | SoftTriple | 89.86% | 89.95% | 86.27% |

Table 3: Average Runtime Statistics (in seconds)

| Dataset | App | Mislabelled Sample | |
|---|---|---|---|
| | | Detection | Recommendation |
| CUB200 | EIF | 260.20 ± 29.28 | 1.85 ± 0.01 |
| | IF | 375.56 ± 68.06 | / |
| CARS196 | EIF | 453.61 ± 28.76 | 3.74 ± 0.03 |
| | IF | 588.11 ± 23.23 | / |
| InShop | EIF | 1306.43 ± 27.31 | 43.24 ± 0.11 |
| | IF | 2454.11 ± 9.55 | / |

evaluate the identified influential training samples (either helpful or harmful) by comparing :

$$\Delta d(p_k) = d(p_k; \boldsymbol{\theta}') - d(p_k; \boldsymbol{\theta}_0) \tag{7}$$

In Equation 7, $\boldsymbol{\theta}_0$ represents the original network, $\boldsymbol{\theta}'$ represents the network *actually* trained by down-weighting harmful samples and up-weighting helpful samples for one epoch, and $d(p_k; \boldsymbol{\theta}')$ represents the L2-normalized embedding distance of the confusion pair $p_c$ on $\boldsymbol{\theta}'$. We test the hypothesis $H_1 : \Delta \bar{d}_{EIF}(p_k) > \Delta \bar{d}_{IF}(p_k)$, i.e., the selected influential samples by EIF can de-confuse the confusion pairs in a more significant way than IF on average. We use the paired t-test for the statistical hypothesis testing, with the assumption that the normality assumption can hold when $K >= 30$ [13].

Furthermore, to evaluate whether EIF can work well on groups of confusion pairs, we select the top-$M$ ($M$=10, 15, 20, 25, 30) testing classes with the most generalization errors, denoted as $\mathcal{G}_M = \{c_1, c_2, ..., c_M\}$. We call each $c_m \in \mathcal{G}_M$ as a confusion pair group. We evaluate the change in average distance after retraining the influential training samples for each confusion pair group.

$$\Delta d(c_m) = \frac{1}{|c_m|} \sum_{p_c \in c_m} d(p_c; \boldsymbol{\theta}') - d(p_c; \boldsymbol{\theta}_0) \tag{8}$$

We test the hypothesis $H_1 : \Delta \bar{d}_{EIF}(c_m) > \Delta \bar{d}_{IF}(c_m)$ over $\mathcal{G}_M$. We use non-parametric hypothesis testing method, i.e. Wilcoxon signed-rank test since $M <= 30$.

**Noisy Data Detection Experiment** In the experiment, we evaluate how effective EIF can detect and recommend to relabel the noisy training samples.

**Results: Influential Sample Identification** Table 1 shows the comparison of the score $\Delta d(p)$ between EIF and the original influence function (IF). We can see that EIF outperforms IF on the average improvement on "deconfusing" the confusion pair. Compared to IF, EIF identifies samples with which retraining the model can increase larger distance for the confusion pair. Moreover, the improvement is of statistical significance (all $p$-values are smaller than 0.001 but the group confusion on the InShop dataset). Compared to other datasets, InShop is a few-shot dataset with more noises (i.e., 50K over 8K classes), which may require a larger training batch size for EIF to be more effective.

**Results: Relabelling Recommendation** Figure 2 shows the performance of EIF on detecting 10% noisy data samples on the three datasets. The x-axis represents the ranked training samples regarding

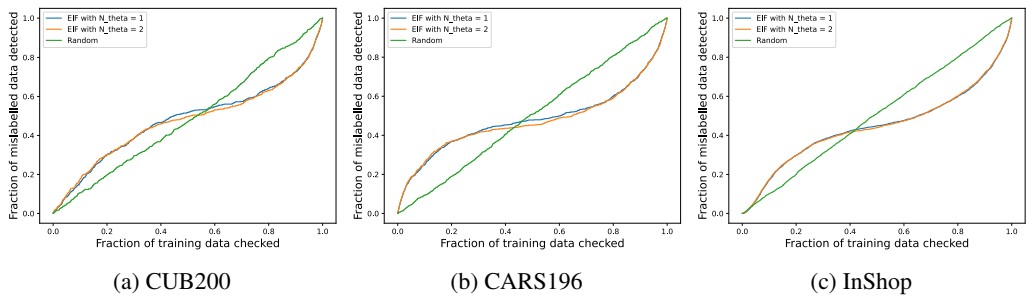

(a) CUB200       (b) CARS196       (c) InShop

Figure 2: The performance of detecting 10% mislabelled samples (see Section 3)

their influence score in descending order (from positive to negative values), the y-axis represents the ratio of mislabelled samples being detected. We plot random choice (in green) and EIF with different $N_{\theta}$ (in blue and orange). Readers can refer to Section 3 for the definition of $N_{\theta}$. We can see that the samples of influence score with small magnitude contribute very few noisy samples; in contrast, either reported harmful or helpful training samples constitute almost all the noisy samples. Our investigation shows that the harmful and the helpful noisy samples are influential in a different way. Comparing to those harmful noisy samples, helpful noisy samples are largely ignored by the model. Thus, they have much larger training loss than the other samples, making them sensitive to model perturbation.

Further, Table 2 shows the performance of recommending the relabelling suggestion. Overall, the sampling relabelling algorithm performs well in predicting and re-calibrating the labels. Note that, with the increase in mislabelling ratio, the relabelling algorithm can still preserve its performance.

**Results: Runtime Performance** Table 3 shows the runtime cost of EIF and IF in the above experiment. We use $N_{\theta} = 1$ for (see Section 4) recording the runtime cost of EIF. In Table 3, we report the mean±std runtime for mislabelled sample experiments. Overall, we can see that EIF can boost the runtime efficiency of IF by ∼33.5% on average.

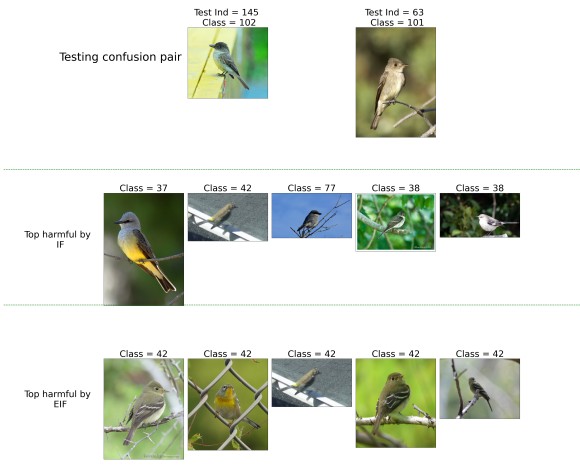

Figure 3: Case study CUB200 example

**Qualitative Result** We qualitatively investigate the top 5 harmful samples identified by EIF v.s. IF. One example is shown in Figure 3. Overall, we have the following two observations (1) EIF identifies top harmful training samples more visually similar to the testing pair (compared to those from IF) and (2) distributed in concentrated classes. More examples can be found in Figure 10, 11, 12, 13, 14.

Table 4: Agreeable and Disagreeable Confusion Pairs

| Dataset | #Confusion | | #Mis-similar | |
|---|---|---|---|---|
| | #Agreeable | #Disagreeable | #Agreeable | #Disagreeable |
| CUB200 | 15 | 5 | 11 | 9 |
| CARS196 | 6 | 14 | 2 | 18 |
| InShop | 10 | 10 | 5 | 15 |

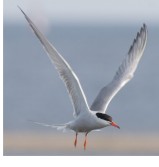
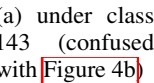
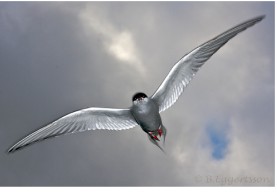
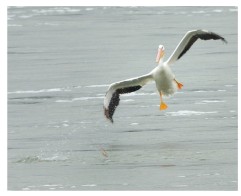
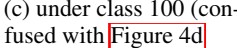
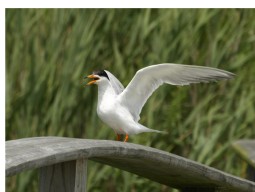

(a) under class 143 (confused with Figure 4b)  (b) under class 140 (confused with Figure 4a)  (c) under class 100 (confused with Figure 4d)  (d) under class 143 (confused with Figure 4c

Figure 4: Figure 4a and Figure 4b are reported as a confusion pair but human investigator agree with the mode decision; Figure 4c and Figure 4d are reported as a confusion pair and human investigator disagree with the mode decision (the birds can be distinguished by the color of their heads).

## 5 Field Study on Popular Datasets

Based on our EIF framework, we further investigate the generalization errors of the state-of-the-art models make on the popular datasets such as CUB200 [37], CARS196 [18], and InShop [19]. We investigate (1) *what does the generalization errors look like and how human agree with the errors?* (2) *what are the root causes for the erroneous model decision?* In this study, we choose Proxy-NCA++ [31] in this study as its leading performance in metric learning community.

**Study Design** For each dataset, we investigate two types of generalization errors: confusion pair and mis-similar pair. The mis-similar pair corresponds to the alternative definition in Section 2, i.e., the semantically similar pair with large distance. For each erroneous pair, we manually evaluate and classify them into: (1) agreeable error and (2) disagreeable error. For agreeable errors, humans agree with the model's decision. For disagreeable errors, humans disagree with the model. In this study, we recruit two volunteers (university graduate students majoring in computer science) to independently verify the reported confusion and mis-similar pairs. For the pairs where they disagree with each other, we let them discuss and reach a consensus. In addition, we use Algorithm 1 to identify and generate relabelling suggestions for the harmful training samples of the erroneous pairs. Based on the recommendation, we further confirm the recommendation and qualitatively analyze the reported problems in training datasets.

**Generalization Errors and Their Agreeability** Table 4 shows that human may share a considerable number of generalization errors with the model. Overall, human agrees with 51.6% (31 out of 60) of the confusion pairs and 30% (18 out of 60) of the mis-similar pairs. We show agreeable and disagreeable confusion pairs in Figure 4. Overall, human investigators agree with many of the "erroneous" model decisions. Readers can check mis-similar pairs on our website [2].

**Root Cause of Erroneous Decision** Table 5 shows that EIF generates relabelling suggestion for 41% (41 out of 100) training classes in CUB200 dataset, 46.9% (46 out of 98) training classes in the CARS196 dataset, and 22.5% (901 out of 3997) training classes in the InShop dataset. Moreover, we further investigate the training classes with more than 10% of their samples recommended to change their labels. We can see that, compared to CUB200 and CARS196, the InShop dataset has more confusing training classes. Figure 5 show some relabelled training samples in the InShop dataset.

We further manually sample the training classes with relabelling recommendations, regarding the following criteria:

- **Centralized Relabelling**: The training classes with more than 10% samples are recommended to be relabelled, and the recommended labels lean towards a single label.

Table 5: Relabelling suggestions on the popular datasets.

| Dataset | #class | #class with relabelling suggestion | | | | |
|---------|--------|-------|------------------------|----------------------|------------------------|--------|
| | | Total | Centralized Relabelling | Diversified Relabelling | Individual Relabelling | Others |
| CUB200 | 100 | 41 | 2 | 3 | 28 | 8 |
| CARS196 | 98 | 46 | 8 | 4 | 26 | 8 |
| InShop | 3997 | 901 | 602 | 201 | 19 | 79 |

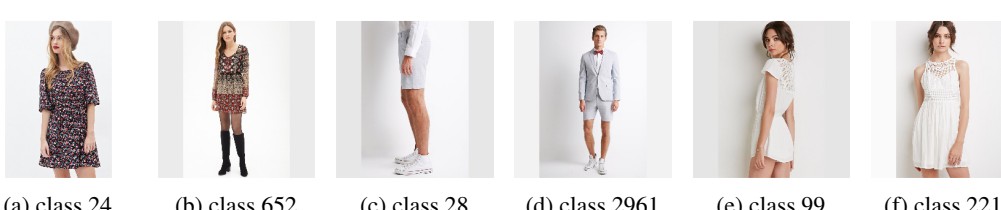

(a) class 24     (b) class 652     (c) class 28     (d) class 2961     (e) class 99     (f) class 2216

Figure 5: Samples in the InShop training dataset, which look similar but under different class. Figure 5a is recommend to relabel to class 652, Figure 5c is recommend to relabel to class 2961, Figure 5e is recommend to relabel to class 2216,

- **Diversified Relabelling**: The training classes with more than 10% samples are recommended to be relabelled, and the recommended labels lean towards diversified labels.

- **Individual Relabelling**: The training classes with less than 5% samples are recommended to be relabelled.

We distinguish centralized and diversified relabelling cases by introducing a threshold $th_H$ (we use 0.35 in this study). Given the entropy of the relabelling-class distribution of a class $\mathcal{C}$ as $H_c$, if $H_c < th_H$, we consider $C$ as a centralized relabelling class; otherwise, we consider $C$ as a diversified relabelling class. Generally, a dataset, if with centralized relabelling requirements, needs to have its relevant training classes merged. In contrast, a dataset, if with diversified and individual relabelling requirements, needs to clean the samples under the relevant training classes.

We sample 3 training classes from each category on each dataset and report the confirmed relabelled suggestions in Table 6. Overall, EIF achieves high recommendation accuracy on the training classes with centralized and diversified relabelling in CUB200 and CARS196; and acceptable accuracy on the training classes with individual relabelling classes in CARS196 and InShop. We provide more details on our website [2].

**Summary and Discussion** In this study, we conclude that the following problems are universal among the popular DML training and testing datasets:

- The testing dataset includes a number of arguably confusing samples, thus an "erroneous" model decision by the labels of testing samples may not necessary be *really* erroneous.

- Some classes are confusing with each other, i.e., over 10% of the samples have the potential to be merged into other classes.

- Many training classes involve outliers that look very different from other samples in the same class, but similar to other classes.

We provide more detailed examples on our website [2]. We conclude that those dataset problems are one of the most important barriers to further improve new state-of-the-art DML approaches. The future work of DML should revolve around dataset cleaning and merging to improve the metrics in a more significant manner.

## 6  Related Work

**Deep Metric Learning** Deep metric learning (DML) learns an embedding space such that intra-class samples are located closer than inter-class samples. Loss functions are usually the key for learning such an embedding, which has been evolved from pairwise-based loss such as [11], [12], [7], and [29] to proxy-based loss such as [20, 31, 6, 34, 8, 23, 14].

Table 6: Manually verified relabelling suggestions on the datasets

| Type | Manual Evaluation | | |
|---|---|---|---|
| | CUB200 | CARS196 | InShop |
| Centralized Relabelling | 100% | 100% | 66.67% |
| Diversified Relabelling | 92.59% | 100% | 50.00% |
| Individual Relabelling | 66.67% | 85.71% | 80.00% |

While new approaches emerge to outperform the state-of-the-art with marginal improvements [21, 10, 24, 40], few work has been proposed towards understanding of generalization error in DML, from the dataset perspective. This work makes the first step, and our findings shed light on the potential problems of the datasets. Moreover, our EIF technique can further facilitate their fixes.

**Model Explanation**   Despite its great successes in multiple disciplines, the deep learning model has remained to be black-box mystery for decades. There are two types of explanations: feature-level and instance-level. While feature-level explanations would like to interpret the semantics and importance of features, instance-level explanations would like to quantify the individual training sample's contribution to prediction. In [15], the idea of influence function is introduced to measure the change in testing loss upon removal of certain training sample. Variations of influence functions are later developed to solve the overestimation for outliers [3], the low diversity in high-influence points [4], the high computaional cost in Hessian estimation [26, 22]. RPS [39] proposes an alternative view from the Represener Point Theorem: they use the weighted kernels of training points as the influence measure. Since RPS has restrictions on model regularizers, RPS-LJE [30] has been later proposed to generalize RPS to models without regularization. However, its definition has subtle differences to the original influence function [15].

Influence function has been applied to various tasks such as VAE [17], GAN [32], data poisoning [9], causal inference [1], data subsampling [33, 36, 35], data relabelling [16]. As far as we know, we are the first work which designs influence function catering to DML problems.

## 7   Conclusion

In this work, we design an empirical influence function (EIF) to debug and understand the generalization errors in state-of-the-art deep metric learning models. Comparing to the traditional influence function, EIF can (1) guide us to locate the influential harmful and helpful training samples and (2) recommend the potential relabelling suggestion for the harmful training samples. Our extensive experiments have proved its effectiveness. With the support of EIF, we further identify the problems of existing datasets for metric learning, which suggests the improvement of the dataset for achieving further world-record performance.

## Acknowledgement

This research is supported in part by the Minister of Education, Singapore (T1-251RES1901, T2EP20120-0019, MOET32020-0004), NUS-NCS Joint Laboratory for Cyber Security, Singapore, the National Research Foundation, Singapore, and Cyber Security Agency of Singapore under its National Cybersecurity Research and Development Programme (Award No. NRF-NCR_TAU_2021-0002) and A*STAR, CISCO Systems (USA) Pte. Ltd and National University of Singapore under its Cisco-NUS Accelerated Digital Economy Corporate Laboratory (Award I21001E0002).

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
