# OpenReview forum: "Debugging and Explaining Metric Learning Approaches: An Influence Function Based Perspective"
_NeurIPS.cc/2022/Conference — NeurIPS 2022 Accept_

### Official Review · Reviewer_7LQB · 2022-07-06

**Rating:** 6
**Confidence:** 4
**Soundness:** 3 good
**Presentation:** 2 fair
**Contribution:** 3 good

**Summary:**


The authors propose EIF, which is a modification of IF, for DML.  Specifically, it differs from IF in two key ways:
- It doesn't require computing the Hessian, which leads to faster calculations.
- It allows for the possibility that the "retrained" parameters may be far away from the original parameters, which is useful when a large portion of the dataset is being re-weighted.

They argue for the efficacy of this method with several sets of results based on datasets with deliberately mislabeled points:
- Showing that the confusion of the most confused pairs  decreases more (statistically significantly) when the model is retrained on points reweighed by EIF than IF (Table 1)
- Showing that EIF is able to identify those mislabeled points (Figure 2)
- Showing that EIF+KNN is able to correctly relabel those mislabeled points (Table 2)

They show that this has computational benefits compared to IF (Table 3)

Additionally, they argue for the efficacy of this method on benchmark tasks:
-  Showing that people often agree with the model's mistakes and, as a result, that the problem is the data labeling (Table 4)
- Analyzing how often EIF identifies different types of data labeling errors

**Questions:**

How should readers compare/interpret the results in Table 1 and Figure 2?
- Table 1 shows that, for the most confused pairs, EIF is better able reduce their confusion than IF.  However, EIF only really appears to beat IF on CUB200 in Figure 2.  What is causing the disagreement between these results?
- Relatedly, all of the influence estimators show a high recovery rate when the fraction of the dataset checked is near 0 or near 1.  Were the influences scores sorted by absolute value?

Doesn't merging similar classes simply make the problem easier?  Fundamentally, it seems that a good model should be able to differentiate similar classes.  So this doesn't seem like a dataset problem that we should try to "fix".

**Limitations:**

Yes

**Strengths And Weaknesses:**

Originality:
- As far as I know, IF has not been applied to DML before
- EIF is, technically, a novel extension of IF

Quality:  Overall, the methodology seems solid
- Having 2 students check the authors conclusions for Table 4 is a little unusual
- Where is the claim in Line 244 supported?

Clarity:
- Line 70-71, it's worth clarifying what "L" is here because, as written, it sounds a lot like you are going to be doing normal supervised learning.
- Equations (1) and (2) are somewhat difficult to parse.  My understanding is that the objective is to solve the 2nd line of these equations and that the first line is like a constraint.
- Equation (2):  Should "l:X->Y" be "R:X->Y"?
- Line 113:  what are "delta d" and "delta L"?  I'm assuming that they are the difference in these metrics for the new and original parameters
- Relatedly, is "delta L" assumed to be negative?  Otherwise, I'm not seeing the intuition for the interpretation of its sign in Lines 114-115.
- Line 156:  how is the distance normalized?
- Clarifying what it means for EIF+KNN to "recommend a new label" would be helpful (especially for understanding table 2).  My guess is that it is "EIF says a point is harmful and KNN assigns it a label different from the provided one" but I am not certain.

Significance:
- This seems like a potentially important problem
- Beyond the the computational benefits, it isn't overwhelming clear the EIF is more effective than IF (see Questions)
- The claims about problems facing DML datasets don't seem especially insightful (some classes have similar images, classes are sometimes similar, classes often have outliers)

---

> ### Author Response · Authors · 2022-08-02
> **Response to Reviewer 7LQB (Part I)**
>
> Thanks to the reviewer for his or her suggestions. We will improve the clarity in our revision.
>
> # Q1: Table 1 shows that EIF is better able to reduce their confusion than IF. However, EIF only appears to beat IF on CUB200 in Figure 2. What is causing the disagreement between these results?
> Thanks for this question. In the DML training experiment, we report influential harmful/helpful training samples to deconfuse the confusing testing pairs; in the noisy experiment, we report influential harmful training samples to *estimate* the noisy samples. That is where the disagreement lies.
>
> Note that the influential harmful samples and the noisy samples are generally correlated, but they are not equal. Note that, in our dataset, the influential harmful samples consist of (i) noisy samples and (ii) the “clean” samples with mislabelling potential (see our field study in Section 5). Moreover, we cannot guarantee that every injected noisy sample is influential to confuse the testing samples. Therefore, EIF can ignore some noisy samples if they are not as influential as some clean samples with mislabelling potential.
>
> # Q2: All of the influence estimators show a high recovery rate when the fraction of the dataset checked is near 0 or near 1. Were the influence scores sorted by absolute value?
>
> No, they are sorted by their signed values. For EIF, negative scores indicate helpful samples while positive scores indicate harmful samples. We use influential harmful samples to estimate noisy samples. In Figure 2, the more left in the horizontal axis, the more harmful samples EIF/IF predicts; and vice versa.
>
> # Q3: Doesn't merging similar classes simply make the problem easier? Fundamentally, it seems that a good model should be able to differentiate similar classes. So this doesn't seem like a dataset problem that we should try to "fix".
>
> We agree that a good model should be able to differentiate similar (*but still different*) classes. However, in the field study, we observe that even humans find it very hard to distinguish the training samples in the same classes (see examples in Figure 3). This observation sheds some light (or at least some arguments) on how to further develop the DML community.
> For example, it is arguable whether we need to further design new DML models based on existing popular datasets, or design a new dataset cleaning framework to fix and even expand those datasets.
>
> # Q4: Having 2 students check the authors conclusions for Table 4 is a little unusual; Where is the claim in Line 244 supported?
> The field study requires participants to manually investigate the reported results. To avoid the bias, the authors are excused to serve as the participants. In the future work, we will recruit more participants to further evaluate the results.
> Our claim is that some training samples are (1) dissimilar to the samples in the same class, but (2) similar to the samples in the other classes.
> We have such a claim based on the fact that the number of the training classes under the category of “individual relabelling” is large. More specifically, a training class in the category has less 5% samples recommended to be relabelled.
> In this experiment, 28 out of 41 classes with relabelling suggestions in CUB200 dataset (26 out of 46 in CARS196 dataset and 19 out of 901 in InShop dataset) fall in this category. We will update Table 5 as follows in our revision. We thank the reviewer’s comment to help clarify our revision.
> | Dataset  | #class | #class with relabelling suggestion |                         |                         |                        |        |
> |----------|--------|:----------------------------------:|-------------------------|-------------------------|------------------------|--------|
> |          |        | total                              | Centralized Relabelling | Diversified Relabelling | Individual Relabelling | Others |
> | CUB200   | 100    | 41                                 | 2                       | 3                       | 28                     | 8      |
> | CARS196  | 98     | 46                                 | 8                       | 4                       | 26                     | 8      |
> | InShop   | 3997   | 901                                | 602                     | 201                     | 19                     | 79     |

---

> > ### Author Response · Authors · 2022-08-02
> > **Response to Reviewer 7LQB (Part II)**
> >
> > # Q5: Clarity issues
> > Thanks for the suggestions, please check our response as follows. We will clarify those points in our revision:
> >
> > # Q5-1: Line 70-71, it's worth clarifying what "L" is here because, as written, it sounds a lot like you are going to be doing normal supervised learning.
> > $L$ refers to the DML training loss in specific.
> >
> > # Q5-2: Equations (1) and (2) are somewhat difficult to parse. My understanding is that the objective is to solve the 2nd line of these equations and that the first line is like a constraint.
> > Yes, we will improve the presentation.
> >
> > # Q5-3: Equation (2): Should "l:X->Y" be "R:X->Y"?
> > Yes, we will improve the presentation.
> >
> > # Q5-4: Line 113: what are "delta d" and "delta L"? I'm assuming that they are the difference in these metrics for the new and original parameters
> > Yes, we will improve the presentation.
> >
> > # Q5-5: Relatedly, is "delta L" assumed to be negative? Otherwise, I'm not seeing the intuition for the interpretation of its sign in Lines 114-115.
> > The sign of $\triangle L(x_{train}; \theta)$ can be either negative or positive, the sign of the product $\triangle d(P_c; \theta) \triangle L(x_{train}; \theta)$ matters. If $\triangle d(P_c; \theta) \triangle L(x_{train}; \theta) < 0$, which means the reduction of the loss of $x_{train}$ (i.e. $\triangle L(x_{train}; \theta)<0$) leads to the increase of the distance of the pair (i.e., $\triangle d(P_c; \theta)>0$), x is helpful in deconfusion. Otherwise, $\triangle d(P_c; \theta) \triangle L(x_{train}; \theta) > 0$ indicates that x is harmful.
> >
> > # Q5-6: Line 156: how is the distance normalized?
> > The embeddings are L2-normalized, and then we take the distances between the embeddings.
> >
> > # Q5-7: Clarifying what it means for EIF+KNN to "recommend a new label" would be helpful (especially for understanding table 2). My guess is that it is "EIF says a point is harmful and KNN assigns it a label different from the provided one" but I am not certain.
> > Yes, we will improve the presentation.

---

> ### Comment · Reviewer_7LQB · 2022-08-05
> **Additional Clarifications**
>
> Q1:  My understanding is that Table 1 is the "main result" while Figure 2 is an explanation of "why that result happens".  Is that correct?
>
> Q2:  Then why are so many of the noisy samples (which presumably should be harmful) found only as almost the entire dataset is checked?  To me, the shape of this curve indicates that noisy samples have high absolute values according to both EIF and IF.  Am I missing something?

---

> > ### Author Response · Authors · 2022-08-07
> > **Response to Additional Clarification by Reviewer 7LQB**
> >
> >
> > > Q1: My understanding is that Table 1 is the "main result" while Figure 2 is an explanation of "why that result happens". Is that correct?
> >
> > No, Table 1 and Figure 2 are two different experiments. Specifically, Table 1 (DML Training Experiment) evaluates the effectiveness of EIF and IF to identify the influential training samples. In contrast, Figure 2 (Noisy Data Detection Experiment) evaluates the effectiveness of EIF and IF to locate the noisy samples in the training dataset. We provide a more detailed explanation as follows.
> >
> > ## - Table 1 (DML Training Experiment)
> > Given a confusion pair or a group of confusion pairs, we use EIF and IF, respectively, to locate the influential helpful and harmful training on the training dataset. Then we verify the influence by retraining the model by up-weighting the helpful samples and down-weighting the harmful samples.
> >
> > We regard that EIF outperforms IF if the model trained with its up-weighted/down-weighted samples reported by EIF can deconfuse the confusion pair(s) more than that of IF, and vice versa.
> >
> > ## - Figure 2 (Noisy Data Detection Experiment)
> > Given a confusion pair or a group of confusion pairs, we use EIF and IF, respectively, to report the noisy samples estimated by the influential harmful training samples. In the experiment, we inject noisy samples in the training dataset. Since both EIF and IF can generate an influence score to indicate harmfulness/helpfulness, we rank the samples by their influence score. For EIF, the larger positive score indicates more harmful sample, the larger negative score indicates more helpful sample, and a score close to 0 indicates the sample with small influence.
> >
> >
> > > Q2: Then why are so many of the noisy samples (which presumably should be harmful) found only as almost the entire dataset is checked? To me, the shape of this curve indicates that noisy samples have high absolute values according to both EIF and IF. Am I missing something?
> >
> > We agree with the reviewer that there is a high correlation between influential training samples (i.e., those with `high absolute value`) and the noisy samples. If we sort the samples by the absolute value of their influential score, EIF and IF can report 72.41\% and 72.41\% of the noisy samples in the first 40\% samples in the CUB200 dataset. The statistics are 87.5\% and 90\% in CARS196 and 81.01\% and 79.07\% in InShop. We thank the reviewer’s comment and will clarify it in our revision.
> >
> > As for the reviewer’s question on why noisy samples can be influential helpful samples, we report our observation as follows. In the DML datasets, some training classes are more similar than the others. For example, In CUB200, the class `Brewer Blackbird` is similar to the class `Red Winged Blackbird`, if the label of the samples are changed into a similar label, those noisy samples can serve as more influential helpful samples to improve the performance.
> >
> > Note that, existing DML community focuses more on improving the metrics while neglecting the *nature* of the dataset. This work serves as a call for such attention.

---

> > > ### Comment · Reviewer_7LQB · 2022-08-08
> > > **Thank you for the clarifications**
> > >
> > > While I'm still not convinced about the usefulness of the observations about how the dataset itself is causing problems (which may simply be because I do not work in DML), I do think that the paper clearly shows that EIF is computationally more efficient than and slightly more effective than IF for DML.  I've updated my score accordingly.

---

### Official Review · Reviewer_5yXY · 2022-07-10

**Rating:** 6
**Confidence:** 3
**Soundness:** 3 good
**Presentation:** 3 good
**Contribution:** 3 good

**Summary:**

The authors propose an empirical influence function which:
(a) Quantifies the impact of training samples on generalization errors in deep metric learning and proposes a relabeling framework in the context of DML.
(b) Proposes a faster version of the traditional influence function which is predominantly slow.

**Questions:**

Please refer to my questions in the Weaknesses section.



**Limitations:**

I feel the paper is a good contribution, though the technical contribution is a bit limited. However, I still feel that the paper should be accepted due to : (A) Application of IF in a new domain such as DML; (b) Good coverage of experiments; (c) Simple strategy to scale up influence functions.

**Strengths And Weaknesses:**

Strengths:

(1) The work is very significant practically as there exists very few works concerning understanding influence functions in contexts other than supervised learning (e.g., DML). To the best of my knowledge, this is one of the only works which investigate model interpretability through the lens of the training data in DML.

(2) While influence function is quite useful for downstream tasks such as relabeling, it is very expensive to compute. The work is also significant in the sense that they circumvent the expensive inverse-Hessian-vector product approximation which is a bottleneck in influence computation.

(3) I have reviewed an earlier version of this paper in another conference. This draft is very well organized and easy to follow.  The authors have also updated the list of references which was inadequate in the earlier draft.

(4) The authors cover a wide range of experiments on most of the commonly used datasets in DML. The empirical analysis supports the results well.

Weaknesses:
(1) While speeding up the influence function via a light weight Newton step approximation, it is not technically novel. However it's still impactful as often it reduces the running time by 10-20 times at the least.

(2)  The authors primary focus on the DML setup and it is fair to not be able to cover different areas of applications. However it would strengthen the paper on how the EIF fares in the supervised setup too. A good idea could be to compare the run-times and also compare the quality of influence values obtained. Considering one of the major contribution is the fast version of IF, it would be beneficial to observe its universality.

(3) Does the selection of backbone architectures play a role in determining the quality of influence values?

---

> ### Author Response · Authors · 2022-08-02
> **Response to Reviewer 5yXY**
>
> Thanks to the reviewer for his or her suggestions.
>
> # Q1: While speeding up the influence function via a lightweight Newton step approximation, it is not technically novel. However it's still impactful as often it reduces the running time by 10-20 times at the least.
> We thank the reviewer’s comment on the impact of our work.
> We also agree that EIF is a less sophisticated solution, compared to IF. However, we regard its simplicity as a strength instead of a weakness. Note that the calculation of EIF is only based on the co-change between the loss of a testing sample and that of a training sample. Therefore, EIF is very general and practical, which makes it much easier to be implemented and adapted in new models such as transformers, and even the graph neural network models.
>
>
> # Q2: It is fair to not be able to cover different areas of applications. However it would strengthen the paper on how the EIF fares in the supervised setup too. A good idea could be to compare the run-times and also compare the quality of influence values obtained. Considering one of the major contributions is the fast version of IF, it would be beneficial to observe its universality.
>
> Thanks for the valuable suggestion. We agree. We will definitely extend our design and experiment on the supervised setup in our future work.
>
> # Q3: Does the selection of backbone architectures play a role in determining the quality of influence values?
>
> Generally, larger model architecture (with more training parameters) can incur runtime overhead to calculate EIF. A mitigation could be to freeze some trainable parameters when we calculate EIF.
> As for how model architecture affects the effectiveness of EIF, we foresee that the influence is small. Within the limited rebuttal period, we further run EIF and IF on the BNInception backbone (a smaller architecture than ResNet-50) on the CUB200 dataset with the ProxyNCA++ loss, to observe their effectiveness to deconfuse the confusing samples. Overall, all the null hypothesis are rejected, and the $H1 : \triangle d_{EIF} (p) > \triangle d_{IF} (p)$ is accepted.
>
> # Various number of confusion pairs
> | number of confusion pairs | average improvement | p-value (paired t-test) |
> |:-------------------------:|:-------------------:|:-----------------------:|
> |             30            |    0.006936666667   |         5.08E-09        |
> |             50            |       0.007648      |         5.45E-14        |
> |             70            |    0.007331428571   |         7.10E-18        |
> |             90            |    0.006923333333   |         7.78E-20        |
> |            100            |       0.006692      |         3.63E-21        |
>
>
> # Various number of groups
> | number of confusion groups | average improvement | p-value (paired Wilcoxon signed rank test) |
> |:--------------------------:|:-------------------:|:------------------------------------------:|
> |             10             |    0.001834897169   |                  0.0009766                 |
> |             15             |    0.001872937321   |                  6.10E-05                  |
> |             20             |    0.001882742691   |                  4.77E-06                  |
> |             25             |    0.001875983859   |                  1.49E-07                  |
> |             30             |    0.00176273209    |                  4.66E-09                  |

---

> > ### Comment · Reviewer_5yXY · 2022-08-08
> > **Response to Authors**
> >
> > Thank you for your response. I would like to maintain my score, as the authors have addressed the questions appropriately.

---

### Official Review · Reviewer_cNAe · 2022-07-26

**Rating:** 5
**Confidence:** 3
**Soundness:** 3 good
**Presentation:** 2 fair
**Contribution:** 3 good

**Summary:**

This paper tackles the question of why Deep Metric Learning (DML) approaches are unable to make more than incremental improvements on classical datasets and tasks. Specifically, the trained models fail to recognize similar samples and tend to make the same kind of generalization errors. To better understand the problem, the authors examine the particular training examples that contribute to the generalization errors that the model makes, and discover that many of these examples are labelled incorrectly. This leads the authors to conclude that the major barriers for DML performance is not model design, but rather the significant presence of incorrectly labelled examples in the classical datasets. To address this issue, they propose an empirical influence function (EIF), which identifies these examples and quantifies their contribution to the generalization error. Moreover, they propose a relabelling technique based on the EIF for the problematic examples.

**Questions:**

My main questions are around the improvement that EIF provides over IF.

In the DML training experiment, it would be interesting to know how the statistics describing the improvement of EIF over IF change as the number of confusing pairs considered is varies. Table 1 of the manuscript reports the statistics when the 50 most confusing pairs are considered - but what happens if the number of pairs is increased or decrease? The statistics for each dataset could be plotted as a function of the number of pairs considered. Similarly the number of confusion groups could be varied from the value 10 for which statistics are currently reported.

Lines 167-168 states 'Compared to IF, EIF identifies samples with which retraining the model can increase larger distance for the confusion pair.'. To make this achievement more concrete for the reader, it would be helpful for the authors to include some examples, perhaps in the appendix, and to refer to these examples at this point in the text. There are some examples of potential mislabelling given in the appendix, but it is not clear how these examples were identified (I couldn't find information about them in the appendix or the main text).

I understand that removing the need to calculate the Hessian greatly improves the runtime over IF, but it would be helpful if the authors can provide some intuition for why the accuracy also improves for the individual pairs. What is the dependence on the choice of N_theta for the DML training experiment, and why in Figure 2 is N_theta relabelled as M?

In Figure 2, for the noisy relabelling experiment, please could the authors comment on why the performance of EIF and IF saturate for the middle tranche of training examples, before rapidly increasing as the last few training examples are checked?

The field study should be carried out for the IF pairs in addition to the EIF pairs, so that Table 4 can be augmented with this data.

Overall there are a few places where the writing could be improved - the meaning is generally more or less clear, but the paper would benefit from a close read though - in some places words or even parameters/variables appear to be missing from sentences, and have to be inferred by the reader from context.

**Limitations:**

It would be helpful if the authors could include a discussion of the limitations of their work.

**Strengths And Weaknesses:**

This is an interesting paper - identifying examples that are mislabelled in large datasets is clearly an important challenge. Figure 2a-c of the appendix is quite compelling evidence for the claims made about these well known datasets in the introduction, and it might be worth adding these panels to the main text.

The authors propose a novel empirical influence function, that tackles the significant run time cost of the existing influence function (IF) and an issue that the authors identify with the accuracy of an approximation made for the calculation of group-pair confusion. I am not an expert in the area of influence functions, but to my knowledge this is a novel proposal and its efficacy is supported by the experimental results that the authors present in the paper.

Identifying mislabelled examples that prevent performance on benchmark tasks from improving is clearly a significant service to the community.

---

> ### Author Response · Authors · 2022-08-02
> **Response to Reviewer cNAe (Part I)**
>
> # Q1: Add statistical significance testing on EIF’s advantage over IF on more various numbers of confusion pairs, and more various numbers of confusion groups.
> To address this suggestion, we add the experiments with the number of pairs = 30, 50, 70, 90, 100 on two datasets CUB200, CARS196 and InShop; and the number of groups = 10, 15, 20, 25, 30. We compare EIF and IF on the ProxyNCA++ loss. Please check the following tables for more details. Overall, all the null hypothesis except for InShop test on group of confusion are rejected, and the $H1 : \triangle d_{EIF} (p) > \triangle d_{IF} (p)$ is accepted.
>
> ## Various number of pairs
> CUB200:
> | number of confusion pairs | average improvement | p-value (paired t-test) |
> |:-------------------------:|:-------------------:|:-----------------------:|
> |             30            |       0.02287       |         3.47E-12        |
> |             50            |       0.021218      |         1.35E-15        |
> |             70            |    0.01993428571    |         4.02E-19        |
> |             90            |    0.01983666667    |         1.88E-24        |
> |            100            |       0.019581      |         5.25E-27        |
>
> CARS196:
> | number of confusion pairs |  average diff | p-value (paired t-test) |
> |:-------------------------:|:-------------:|:-----------------------:|
> |             30            |    0.01473    |         8.16E-06        |
> |             50            |    0.012008   |         1.04E-07        |
> |             70            | 0.01311857143 |         3.06E-11        |
> |             90            | 0.01249666667 |         1.52E-12        |
> |            100            |    0.012729   |         2.93E-13        |
>
>
> InShop:
> | number of confusion pairs |  average diff | p-value (paired t-test) |
> |:-------------------------:|:-------------:|:-----------------------:|
> |             30            | 0.01519666667 |         1.96E-05        |
> |             50            |    0.013948   |         2.98E-07        |
> |             70            | 0.01594714286 |         7.76E-11        |
> |             90            | 0.01603222222 |         2.10E-13        |
> |            100            |    0.015058   |         9.27E-14        |
>
> ## Various number of groups
> CUB200:
> | number of confusion groups | average improvement | p-value (paired Wilcoxon signed rank test) |
> |:--------------------------:|:-------------------:|:------------------------------------------:|
> |             10             |    0.02226814021    |                  0.0009766                 |
> |             15             |     0.0234228876    |                  3.05E-05                  |
> |             20             |    0.02427592818    |                  9.54E-07                  |
> |             25             |    0.02435943452    |                  2.98E-08                  |
> |             30             |    0.02435346258    |                  9.31E-10                  |
>
>
> CARS196:
> | number of confusion groups | average improvement | p-value (paired Wilcoxon signed rank test) |
> |:--------------------------:|:-------------------:|:------------------------------------------:|
> |             10             |    0.01871259603    |                  0.0009766                 |
> |             15             |    0.01950539567    |                  3.05E-05                  |
> |             20             |    0.01828522803    |                  9.54E-07                  |
> |             25             |    0.01724989733    |                  2.98E-08                  |
> |             30             |    0.01679854478    |                  9.31E-10                  |
>
> InShop:
> | number of confusion groups | average improvement | p-value (paired Wilcoxon signed rank test) |
> |:--------------------------:|:-------------------:|:------------------------------------------:|
> |             10             |   -0.0007832444494  |                  9.35E-01                  |
> |             15             |   0.0006221088001   |                  7.38E-01                  |
> |             20             |   0.0004720652132   |                  7.63E-01                  |
> |             25             |    0.002077030729   |                  5.11E-01                  |
> |             30             |    0.003266685399   |                  3.35E-01                  |
>
> # Q2-1: Find more examples to support the claim ``EIF identifies samples with which retraining the model can increase larger distance for the confusion pair’’.
> Thanks for the suggestion. We prepare more examples at our anonymous website URL: https://sites.google.com/view/empirical-influence-function/case-study-on-de-confusion-ability-for-eif-vs-if, we will include them in our revision.
> Visually, given a pair of confusing testing samples, compared to IF, EIF identifies top harmful training samples (1) more visually similar to the testing samples and (2) distributed in concentrated classes;

---

> > ### Author Response · Authors · 2022-08-02
> > **Response to Reviewer cNAe (Part II)**
> >
> > # Q2-2: There are some examples of potential mislabelling given in the appendix, but it is not clear how these examples were identified (I couldn't find information about them in the appendix or the main text).
> >
> > We identify the samples with mislabelling potential with the following steps:
> > - Step 1: We use EIF to select harmful training samples by tracing the most influential training from top 10 frequently confused testing classes.
> > - Step 2: We use EIF to recommend the new labels for those harmful training
> > - Step 3: If manual investigation can confirm the relabelling recommendation, we present those samples as a part of qualitative analysis
> > We will clarify this in our revision.
> >
> > # Q3-1: provide some intuition for why the accuracy of EIF is improved over that of IF for the individual pairs.
> > IF and EIF use different measurements to calculate influence function with the same rationale. Without losing generality, we use $L(x_{test})$ to represent the confusion distance $d(P_c)$ in the paper. Specifically, IF calculates the derivative of the test loss over the changed weight of any training samples (i.e., $\frac{\partial L(x_{test})}{\partial \epsilon}$). In contrast, given the existing model $\hat\theta$, EIF calculates the expectation of training-testing loss co-change  (i.e., $E_{\theta’}(cop(\triangle L(x_{test}) \triangle L(x_{train})))$), where we sample a number of $\theta’$ close to $\hat\theta$.
> > Technically, compared to IF, EIF has less estimation steps in practice, which can allow EIF sometimes to be more effective.
> >
> > ## Estimation in IF
> > - The induction of the IF formula (i.e., $\frac{\partial L(x_{test})}{\partial \epsilon} = -\nabla L(x_{test}; \hat\theta)^T H_{\hat\theta}^{-1} \nabla L(x_{train}; \hat\theta) $) assumes that (i) the model parameter $\hat\theta$ is at the saddle point, and (ii) the empirical risk is twice-differentiable and strictly convex. However, these may not hold in practice. Therefore, to make the IF solution feasible in practice, IF needs to
> > (1) estimate $H_{\hat\theta}^{-1} \nabla L(x_{test}; \hat\theta)$ via stochastic estimation
> > (2) estimate non-convexity loss with a convex quadratic loss
> > (3) estimate non-twice-differentiable loss with smooth approximation.
> > Despite that the IF solution has empirically proved to be effective, each step incurs estimation errors, which cannot guarantee the result is as effective as the theory suggests.
> >
> > ## Estimation in EIF
> > - In EIF, when estimating $E_{\theta’}(cop(\triangle L(x_{test}) \triangle L(x_{train})))$, we have only ONE estimation step, i.e., sampling $\theta$’s with the directions to maximize/minimize $L(x_{test})$. It is because these two directions allow us to observe the co-change of $\triangle L(x_{train})$ and $\triangle L(x_{test})$ in the most effective manner. Thus, they are among the most informative $\theta$’s to sample.
> >
> > Our experiment shows that EIF outperforms IF in deconfusing testing samples, while EIF and IF are comparable in noisy data detection.
> >
> > # Q3-2: What is the dependence on the choice of N_theta for the DML training experiment, and why in Figure 2 is N_theta relabelled as M?
> > Thanks for pointing out this. We use $N_{\theta}=1$ in the DML training experiment and will make $N_{\theta}$ and $M$ consistent in our revision.
> >
> > # Q4: In Figure 2, why does the performance of EIF and IF saturate for the middle tranche of training examples, before rapidly increasing as the last few training examples are checked?
> > Both EIF and IF report signed influence scores. For EIF, the more positive the score, the more harmful it is, and vice versa. In Figure 2, the more left in the horizontal axis, the more harmful samples EIF/IF predicts; and vice versa. Thus, the saturation plateau are the samples with minimum influence (i.e., smallest absolute value).
> > We use reported harmful samples to estimate the noisy samples. Overall, EIF and IF have comparable performance to identify the noisy samples. Noteworthy, EIF takes 33.5% less time than IF.
> >
> > # Q5: The field study should be carried out for the IF pairs in addition to the EIF pairs, so that Table 4 can be augmented with this data.
> >
> > Compared to IF which only predicts the influence score, EIF also recommends new labels for the harmful samples. Since the design of our field study also targets for the *fix* of the samples with mislabelling potential (see “Root Cause of Erroneous Decision” in Section 5), we regard EIF as a more appropriate solution. We will extend the field study with IF-based reports in our future work.
> >
> > # Q6: Overall there are a few places where the writing could be improved
> > Thanks for the suggestions, we will thoroughly check our paper to fix this comment.

---

### Meta-Review · Area_Chair_Fq2X · 2022-08-26

**Recommendation:** Accept
**Confidence:** Certain

**Metareview:**

This paper approaches the problem of debugging failures in deep metric learning, developing and applying a more efficient version of influence functions called EIF (empirical influence functions) that is shown to effectively help root-cause failures to ambiguous or poorly labeled examples in standard training sets.

The reviewers found this paper to be deeply interesting, appreciating the efficiency but moreover the novelty of applying IF-style approaches to the problem of DML.  From my perspective, I think the novelty point is even stronger than some of the reviewers called out -- yes, it there is some novelty in the development of EIF, but the real novelty is in creating and evaluating effective debugging strategies for learned metric spaces.  This kind of analysis and debugging is sorely lacking in the overall literature, and I am very happy to see this work present a compelling and intuitive approach -- both for its own merit, and also for the similar ideas it may well inspire in adjacent areas of research.

**Award:**

No

---

### Decision · Program_Chairs · 2022-09-14

Accept